# Don't bite off more than you can chew: Investigating Excessive Permission Requests in Trigger-Action Integrations

## ABSTRACT

Various web-based trigger-action platforms (TAPs) enable users to integrate diverse Internet of Things (IoT) systems and online services into trigger-action integrations (TAIs), designed to facilitate the functionality-rich automation tasks called *applets*. A typical TAI involves at least three cooperative entities, i.e., the TAP, and the participating trigger and action service providers. This multi-party nature, nonetheless, can render the integration susceptible to security and privacy challenges. Issues such as risky action mis-triggering and sensitive data leakage have been continuously reported from existing applets by recent studies.

In this work, we investigate the *cross-entity* permission management in TAIs, addressing the root causes of the applet-level security and privacy issues that have been the focus of the literature in this area. We advocate the *permission-functionality consistency*, aiming to reclaim *fairness* when the user is requested for permissions. We develop PFCon, which extracts the required permissions based on all functionalities offered by an entity, and checks the consistency between the required and requested permissions on users' assets. PFCon is featured in leveraging advanced GPT-based language models to address the challenge in the TAI context that the textual artifacts are short and written in an unformatted manner. We conduct a large-scale study on all TAIs built around IFTTT, the most popular TAP. Our study unveils that nearly one third of the services in these integrations request excessive permissions. Our findings raise an alert to all service providers involved in TAIs, and encourage them to enforce the permission-functionality consistency.

## 1 INTRODUCTION

Web-based Trigger-Action Platforms (TAPs) like IFTTT [2] and Zapier [5] have enabled the integration of IoT systems with a great variety of online services, ranging from cloud applications and development tools to social media. These platforms facilitate what we term as *Trigger-Action Integrations* (TAIs), enabling cross-party data and control flows to execute if-then style automation tasks, or *applets*. For instance, applets such as "*IF a new email arrives, THEN sync the attachment to MyDrive cloud*" can be effortlessly implemented by lay users, obviating the need for programming skills. As of 2023, the popularity of such platforms is evident, where IFTTT alone boasts 20 million registered users and supports 75 million applets [8].

However, this convenience comes at a cost: the multi-party nature of TAIs has inadvertently turned them into a hotbed for security and privacy vulnerabilities. Attackers can exploit unexpected chaining of co-installed applets or manipulate triggers to invoke privileged or unintended actions [38, 42, 43]. Moreover, some TAIs unnecessarily include sensitive data attributes in triggers [13, 18], thereby violating privacy norms. Recent studies [20, 32, 37] have also highlighted that many popular TAIs are non-compliant with existing data protection regulations like the EU's General Data Protection Regulation (GDPR) [1], raising concerns about the permanent storage of user data by TAPs or participating services.

Many efforts have been made by the research community to address these issues, through checking the applet chaining [17, 44], analyzing TAIs [23, 28, 39] and restricting the information collection and information flow [16, 18, 29]. They mainly focus on mitigating security and privacy issues arising upon the applet creation and execution. This unfortunately offers only a partial view of the security and privacy protection in TAIs, as the fundamental cause of these issues is the lack of a comprehensive *cross-entity* data flow and access control among the services inside the TAI.

The TAP connects participating services mostly through the OAuth protocol [4]. Upon the connection time, the user is prompted to grant permissions requested by the TAP on their data and objects managed by the participating services. They have to either accept all listed permissions or decline the integration of the services. Even though the new OAuth 2.0 protocol [3] replaces such an *all-or-nothing* paradigm with a fine-grained permission authorization[1], it remains challenging for lay users to make appropriate selections due to the lack of knowledge on applets' execution workflows. Consequently, most users tend to simply grant all requested permissions, despite their concerns on the insufficiency of security and privacy protection provided by the TAP. The core issue in this permission management is that it is largely in favor of the TAP and service providers rather than the users, as *all behaviors of TAIs can be claimed to be under the coverage of user authorization*.

**Our work**. In this work, we target the problem of *whether the services in a TAI request and/or expose unnecessary permissions beyond the need of their functionalities, particularly, constructing their triggers and/or executing their actions*. We advocate the *permission-functionality consistency*, to address the root causes of the applet-level control and data flow [18, 36] that existing studies mostly focus on. The permission-functionality consistency is twofold, on the *object level* and the *operation level* respectively. First, when a participating service requests to operate on the user's object managed by another service provider, the access scope should be limited to what is necessary in relation to the purposes of the automation task. For example, the action of "*sync the attachment to MyDrive cloud*" needs the access to the attachment only, and that to the email body is unnecessary. Second, when the TAP requests a service provider to undertake an operation, it should comply with the principle of least privilege [34]. For example, the action above requires the *write* permission to the user's cloud folder, but the *read* or *delete* permission is unnecessary.

We propose PFCon, a framework which checks TAI for violations of permission-functionality consistency (or simply *permission excess*). PFCon derives the permissions needed on objects and operations for a participating service to fulfill its functionality, called

---

[1]While developers have the option to include tickable choices in their OAuth prompts, we notice that only four services have actually implemented this particular design.

*required permissions*, and then retrieves the set of permissions that the TAP requests from the user, called *requested permissions*. The particular challenges in this process are three-folds. First, when creating the TAI, the requested permissions are not listed until the OAuth authentication is completed for IFTTT (as the *relying party*) to be registered to the participating services, and most participating services customize the OAuth process (***Challenge #1***). To handle this, we build an authentication and authorization engine to automatically drive the integration of the TAI. Second, the functionality and authorization artifacts are written in natural language, with varying format and quality among services (***Challenge #2***). We resort to the latest large language models (LLMs) to interpret them, and conduct *in-context learning* with *chain of thoughts* to guide them with domain knowledge. Third, the terms used to describe permissions differ among services, causing misalignment for the consistency checking (***Challenge #3***). To address this, PFCon constructs lattice systems to describe the hierarchy among objects and operations, and use them as a context for LLMs to align the terms.

To understand the *status quo* of the permission excess issues in existing real-world TAIs, we conduct a large-scale study using PFCon on all TAIs integrated around IFTTT, with over 700 popular services such as Monzo, Amazon Alexa, Instagram and Facebook. We comprehensively analyze 427 of them that can be executed without special requirements such as IoT devices. We have identified surprisingly prevalent permission excess issues, as 179 services are over-requested by IFTTT with at least one unnecessary permission. In particular, 62 services are over-requested for sensitive operation permissions such as modifying or even deleting users' files and freezing users' credit cards, while 131 are over-requested for object permissions to accessing privacy-sensitive content like videos.

**Contributions**. The main contributions of this work are summarized as follows.

- **Understanding the permission excess issues in TAIs**. We advocate the principle of permission-functionality consistency in TAIs and characterize the permission excess issues, when multi-party services are connected through OAuth.
- **A systematic assessment approach**. We propose PFCon, which implements a series of techniques to automatically identify permission excess issues from TAIs.
- **Revealing the *status quo* and findings of permission excess in real-world TAIs**. Our large-scale study reveals that the permission management in current TAIs is problematic. We also investigate the causes for the permission excess issue. Our findings should raise an alert to the users, and encourage the service providers in TAIs to redesign their interfaces in TAI construction.

## 2 PROBLEM FORMULATION

### 2.1 Background and a Running Example

The TAP and service providers are typically integrated following an OAuth procedure, which includes three phases (shown in Figure 1). We use the action service (i.e., MyDrive) in the example applet "*IF a new email arrives, THEN sync the attachment to MyDrive cloud*" for illustration.

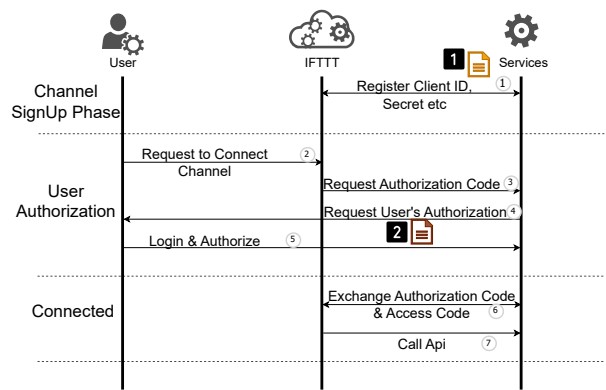

**Figure 1: A general workflow to integrate services into TAP**

**Phase 1: channel sign-up**. IFTTT builds a *channel* for each service it supports (detailed soon in this section). The channel can be regarded as an agreement[2] on the interfaces that IFTTT can use and the service provider should implement (step ① in Figure 1). Then, IFTTT is registered as a trusted client and is assigned a client id.

**Phase 2: user authorization**. When IFTTT is requested by the user to integrate a service, e.g., the MyDrive service, it asks MyDrive to authorize it with the permissions to invoke the APIs defined in the channel on behalf of the user. MyDrive then requests the user to log into their account, and displays a permission prompt (Document ②  in Figure 2) that specifies the permissions requested (step ④ and ⑤). The user can select "*Yes*" to confirm the authorization or "*No*" to cancel it.

**Phase 3: applet execution**. Once the user grants the permissions, IFTTT is given an OAuth access token (step ⑥). It can use the access token as a credential to access the user's account, manipulate objects or perform actions when the applet is executed (step ⑦). Note that the access code bound with the set of permissions specified in Phase 2 and MyDrive checks its validity each time.

**Service functionality artifacts**. During the channel sign-up process, IFTTT provides the participating service providers with comprehensive documents about the interfaces and functionality of the TAI, such as the service name, description, and supported APIs. The Document ①  in Figure 2 shows part of such information in our running example. These documents contain sentences in natural language and formatted or semi-formatted phrases (e.g., an API name of "*new file in folder*"). They describe mainly three types of APIs supported in IFTTT, i.e., trigger APIs, query APIs and action APIs. The former two are for triggers (in *push* model and *pull* model respectively), and the latter is for actions. Table 1 lists some example APIs. We call all these documents and descriptions *service functionality artifacts*, and PFCon targets to infer from them the permissions that are needed to fulfil the defined functionalities.

**Service and Applet**. IFTTT utilizes APIs (Document ①  in Figure 2) from different services to create executable applets. For example, in the applet "*IF a new email arrives, THEN sync the attachment to MyDrive cloud*", it includes a trigger API, "*new_email_arrive*"

---

[2]This means the interfaces are *mutually* defined. Due to this, we treat the entire integration of IFTTT and participating services, i.e., the TAI, as the subject of liability.

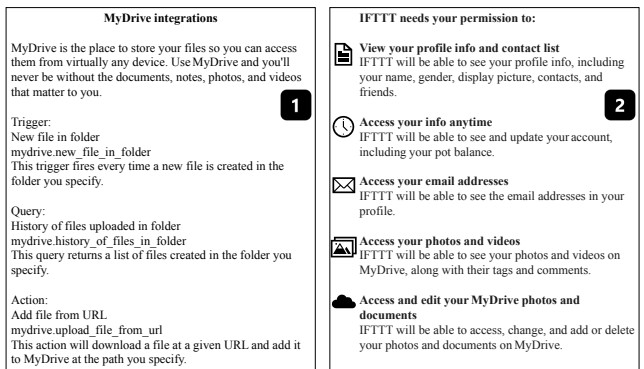

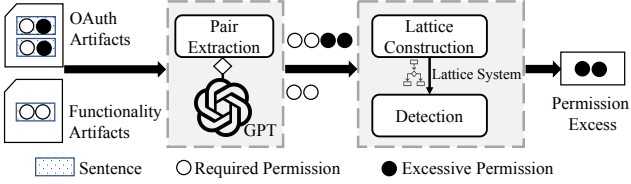

**Figure 2: An example of the IFTTT artifacts**

**Table 1: Trigger, Query and Action APIs in MyDrive**

| Category | API endpoint slug | Parameter | Return value |
|---|---|---|---|
| Trigger | new_file_in_folder | folder_path | name, modified_by |
| | new_photo_in_folder | folder_path | name, modified_time |
| Query | history_of_photo | folder_path | name, modified_time |
| | history_of_files | folder_path | name, modified_by |
| Action | append_to_text_file | filename, content | nil |
| | create_text_file | filename, content | nil |

from the Email Service and an action API, "*upload_a_file*" from the MyDrive Service. One service can provide multiple APIs to IFTTT, allowing for the creation of numerous applets by connecting to other services. As a result, IFTTT offers 59,009 applets [18] based on 700 services.

### 2.2 Threat Model and Scope

**Scope**. PFCᴏɴ targets the permission excess problem of *whether the TAI over-requests functionality-unnecessary permissions*. The core idea of PFCᴏɴ is to extract the required permissions and requested permissions from available artifacts, and then check the inconsistency between them. When a participating service is asked to provide more permissions than what is needed to fulfil its functionalities, in terms of executing all its APIs related to triggers/queries/actions, PFCᴏɴ reports it as a permission excess issue. Since the interfaces between the participating services and IFTTT are mutually established, PFCᴏɴ treats them together, i.e., the TAI, as the liability subject. We explore the permission excess issue from both the object level and the operation level.

- **Object level**. The TAI should comply with limited data restriction. In our running example, the action of "*sync the attachment to cloud*" needs the access to attachment only, rather than the email body.
- **Operation level**. When the TAI undertakes an operation on an object, it should execute the least privilege. In our running example, the action requires the *write* permission to the cloud folder, but not the *read* or *delete* permission.

It is worth noting that PFCᴏɴ focuses on service-level permission management, in contrast to previous studies [9, 14, 18, 19, 23,

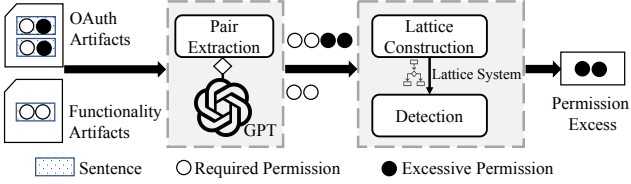

**Figure 3: Overall workflow of PFCᴏɴ**

32, 36, 42, 44] that analyze privacy and security concerns arising with the creation and execution of individual applets.

### 2.3 Permission Excess Definitions

**Definition 1 [Permission]**. Each permission is defined as a pair $P = (Operation, Object)$, denoted by (OP, OB), which means the entity with $P$ possesses the right to perform the $Operation$ on the $Object$.

**Definition 2 [Permission excess]**. We use $\mathcal{S}$ to indicate the set of the requested permissions and $\mathcal{R}$ the set of required permissions for the correct functionality of the service, where $\mathcal{S}$ is derived from the OAuth authorization requests (e.g., Document 2 in Figure 2), and $\mathcal{R}$ is derived from service functionality artifacts (e.g., Table 1 and Document 1 in Figure 2 demonstrating part of the artifacts). A permission excess occurs when any permission requested is not required.

## 3 OUR APPROACH

### 3.1 Overview of PFCᴏɴ

PFCᴏɴ consists of three main components, i.e., *artifact collection*, *permission recognition* and *permission excess detection*, as shown in Figure 3. Below we brief each of them.

**Artifact collection**. This component aims to collect data for inferring (OP, OB) pairs in $\mathcal{S}$ and $\mathcal{R}$. The data needed by PFCᴏɴ includes the OAuth authorization page (for $\mathcal{S}$) and the functionality artifacts (for $\mathcal{R}$). The challenge to automate the authentication process to obtain the authorization page (i.e., ***Challenge #1***). This component is detailed in Section 3.2.

**Permission recognition**. This component aims to recognize the (OP, OB) pairs from the collected artifacts and derive the permissions in $\mathcal{S}$ and $\mathcal{R}$. It has to interpret the artifacts that are written in natural language, with varying format and quality (i.e., ***Challenge #2***). This component is detailed in Section 3.3.

**Permission excess detection**. This component checks the permission inconsistency between $\mathcal{S}$ and $\mathcal{R}$. The challenge lies in the diversity and context-sensitivity of the terminologies in (OP, OB) pairs (i.e., ***Challenge #3***). For example, subsumptive relationships are present among service-level objects and operations. It is crucial to address self-defined structures like "money pot ≺ user account" in online banking appropriately by considering the context. This component is detailed in Section 3.4.

### 3.2 Artifact Collection

PFCᴏɴ collects two types of artifacts, i.e., functionality artifacts (for deriving $\mathcal{R}$) and authorization pages (for deriving $\mathcal{S}$), as illustrated

in Figure 2. First, PFCon crawls the developer site of IFTTT for functionality artifacts. The developer site is an interface first identified by a state-of-the-art TAI analyzer named Taifu [32] using reverse engineering. It provides web APIs[3], which include rich information about the functionality, usage and user statistics (e.g., installation). As it takes service names as inputs, PFCon fetches a full list of participating services supported by IFTTT and passes their names to the web APIs to get all artifacts related to each service.

Obtaining the authorization page turns out to be challenging, given that service providers tend to implement the authentication process in diverse ways with various UI items. To obtain the authorization pages that only appear during authorization, PFCon has to automate the connection between different services and IFTTT, which includes authentication (via user credentials) and authorization (via prompted windows) of OAuth. We create a set of test accounts for PFCon to complete this process. It uses an HTML parser to process each web page and search for all tags that potentially involve user inputs or user interactions, for example, `<input>`, `<button>` and `<a>`. From them, it then identifies the type of each tag, including the username/password fields, and navigation/login/authorization buttons, based on the attributes of the tag, e.g., `id`, `class`, and `name`. The username/password fields are identified from `input` and `div` tags, and login/authorization actions are identified from `a` and `button` tags. To comprehensively capture the permission-related information, PFCon extracts all visible texts throughout the connection process. It also filters out two types of services, i.e., non-English services (merely 4.8% of the entire corpus) and services that do not require OAuth authorizations.

### 3.3 Permission Extraction

With the crawled artifacts, PFCon can proceed to construct the (OP, OB) pairs in $\mathcal{S}$ and $\mathcal{R}$. The challenge arising is that the artifacts are written in natural language, with varying format, terms and quality used among service providers. To address this, we resort to the large language models (LLMs) for interpreting the artifacts, considering that LLMs are capable of assimilating correct syntax and semantics due to their intricate model architecture and extensive training data. We adopt the state-of-the-art GTP-4 model [15], and design a set of prompt patterns for it, as listed in Table 2. To further guide it with domain knowledge, we conduct an *in-context learning* [31] with the *Chain of Thoughts* (CoT) mode [41] applied.

We disintegrate the permission extraction task into two small tasks to for the precision of the LLM, including *sentence separation* and (OP, OB) *extraction*. The former breaks down complex sentences into a series of simpler sentences that contain (OP, OB) pairs. The prompt pattern and the example is shown in the *sentence separation* row in Table 2. Given a complex sentence in the crawled artifacts, e.g., "*access, change, and add or delete your photos and documents*", PFCon requests the LLM to "*assist me in breaking down the following complex sentence into a series of simpler sentences*". CoT prompts are provided to guide the LLM through the *reasoning* process. For example, "*This sentence involves four operations—access, change, add, and delete—on two objects: photos and documents. Therefore, the total number of resulting sentences should be 4 * 2 = 8*". Employing a similar approach, PFCon further extracts the representation of

---

[3]https://ifttt.com/api/v3/graph

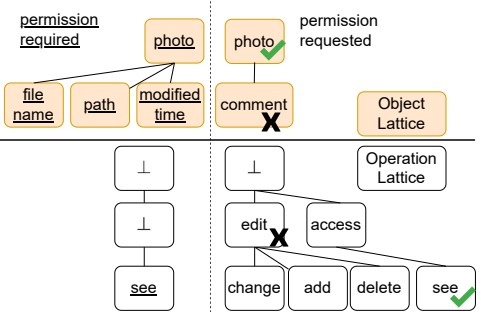

**Figure 4: Permission lattice system of the running example**

(OP, OB) pairs from the obtained simple sentences utilizing the LLM, as shown in the (OP, OB) *extraction* in Table 2.

### 3.4 Permission Excess Detection

Based on the extracted (OP, OB) pairs in $\mathcal{S}$ and $\mathcal{R}$, PFCon detects permission excess issues. Rather than simply comparing the operations and objects, it constructs the permission hierarchy (Section 3.4.1) to enable fine-grained and service context-aware checking, and the detects permission excess based on it (Section 3.4.2).

*3.4.1 Constructing Permission Lattice Systems.* Similar to other systems like mobile OSes [21], most participating services also have a complex permission system. This often leads to a hierarchical structure among the permissions. For example, *"see your photos (including contents, comments, etc.)"* contains higher privilege than *"see comments of your photos"*. To preserve such hierarchical structures for the permission excess checking, we propose the lattice system representation similar to prior studies [24, 37] to facilitate the permission excess checking. PFCon builds two permission lattice systems for each service, i.e., *object lattice* and *operation lattice*. Before diving into details, we show the constructed lattice systems of our running example in Figure 4. As shown in the figure, IFTTT requests the permission (upper right) to access the *"photo"* and its *"comments"*, whereas the permission required (upper left) only includes metadata (i.e., name, folder path, modified time) of the *"photo"* object. PFCon considers *"comments"* as an object permission excess. Similarly, according to the operation lattice system in Figure 4, IFTTT requests the permission (lower right) to perform *"change"*, *"add"*, and *"delete"* operations over a photo object under *"edit"*. However, the required permission (lower left) only includes the access to *"see"* the photo. PFCon considers this as an operation permission excess.

PFCon mainly relies on sentence-level lexicosyntactic patterns and applet data structures to determine the subsumptive relationship between two terms. Below we discuss them.

**Lattice of $\mathcal{R}$ (objects/operations).** We use the relations in the data structures that IFTTT uses to construct the TAI (examples shown in Table 1) to construct the lattices for both objects and operations. When an object/operation is identified from a field, other objects/operations identified from its subfields are considered as subsumed, denoted by $\prec$ (or $\preceq$ if the two may also refer to the same). Take the API *"new_photo_in_folder"* in Table 1 as an example. It contains parameters (e.g., *"folder path"*) and return values (e.g.,

**Table 2: Prompt patterns and tutorials used for four tasks of PFCon**

| Id | Task | Prompt pattern | Input |
|---|---|---|---|
| | | **Permission extraction: sentence separation** | |
| 1 | Tutorial | Break down a complex sentence *<Complex_Sentence>* into a series of *<Simpler_Sentences>*, each comprising only one operation and one object. The simple sentence contains no conjunctions. *<Chain_of_Thought>* | IFTTT will be able to access, change, and add or delete your photos and documents on MyDrive. |
| | | | access your photos; change your photos; access your documents; change your documents; add to your photos; delete your photos; add to your documents; delete your documents. |
| | | | This sentence contains four operations: access, change, add, and delete on two objects: photos and documents. So the total number of result sentences should be 4 * 2 = 8, as follows: |
| | Actual task | Assist me in breaking down the following *<Artifacts_Sentence>* into a series of simple sentences, each comprising only one operation and one object. The simple sentence should contain no conjunctions. | |
| | | **Permission extraction: (OP, OB) extraction** | |
| 2 | Tutorial | The sentence *<Sentence>* contains an Operation-Object pair *<Operation, Object>*. | Access your photos. |
| | | | Operation: Access
Object: your photos. |
| | Actual task | Get the Operation-Object pair from the following sentence *<Artifacts_Sentence>*. | |
| | | **Object-level detection** | |
| 3 | Tutorial | *<Context>*. Is the permission *<Permission>* required by *<Service_API>* based on the object hierarchy *<Service_Lattice_System>*? *<Chain_of_Thought>* So, the answer is Yes. | Under the context of 'MyDrive', the general description of the service function is 'MyDrive' is the place to store your files so you can access them from virtually any device. Use MyDrive and you'll never be without the documents, notes, photos, and videos that matter to you.' |
| | | | Does MyDrive API that can access the object 'pot balance' require the permission to access the object 'account' Lattice System: *pot balance ≺ account* |
| | | | Within the lattice system, the relationship hierarchy is as follows: *pot balance ≺ account*. As a result, there is a subsumptive relationship between 'pot balance' and 'account'. We conclude that access to the pot balance requires permission to access the account. |
| | Actual task | *Context>*. Is the permission *<Permission>* required by *<Service_API>* based on the object hierarchy *<Service_Lattice_System>*? | |
| | | **Operational-level checking** | |
| 4 | Tutorial | *<Context>*. Is the permission *<Permission>* required by *<Service_API>* based on the operation hierarchy *<Service_Lattice_System>*? *<Chain_of_Thought>* So, the answer is No. | Under the context of 'MyDrive'. the general description of the service function is 'MyDrive is the place to store your files so you can access them from virtually any device. Use MyDrive and you'll never be without the documents, notes, photos, and videos that matter to you.' |
| | | | Is permission to 'update your account' required by MyDrive API that can perform the action 'see your pot balance' Lattice System: object *pot balance ≺ account*, the operation on object account: *see ≺ unknown, update ≺ unknown* |
| | | | In the context of the lattice system, it is established that the object *pot balance* is subordinate to *account*, denoted as ≺ (*pot balance ≺ account*). Consequently, when one seeks authorization to 'see your pot balance,' it necessitates permission to 'see your account.' For operations 'see' and 'update' belong to distinct parallel hierarchies, with no inherent subsumptive connection between them. Consequently, the permission 'update your account' is not mandated for an API capable of executing the 'see your account' action. As a result, it can be deduced that the permission 'update your account' is similarly not obligatory for an API authorized to perform the 'see your pot balance' action. |
| | Actual task | *<Context>*. Is the permission *<Permission>* required by *<Service_API>* based on the operation hierarchy *<Service_Lattice_System>*? | |

"name" and "modified time"). We construct the object lattice (upper left of Figure 4) and specify "folder path", "name" and "modified time" as the objects that belong to the object "photo". For operation lattice system under the object of "photo", since "photo" appears in a trigger as "new_photo_in_folder" and a query as "history_of_photo", and no action (shown in Table 1), we can construct the operation lattice that only consists of "see" (lower left of Figure 4).

**Lattice of $\mathcal{S}$ (objects)**. We use lexicosyntatic patterns that are proposed by PolicyLint [10] to capture the subsumptive relationship in a sentence. These patterns define the relationship before and after keywords like "such as", "e.g.|i.e.", "for example" and "include". For example, from the sentence "IFTTT would see your photos, along with their comments", "comments" is found to be one attribute of "photos". Therefore, we can derive that *comments ≤ photos* (see upper right of Figure 4). We expand the patterns of PolicyLint with additional keywords to enhance its accuracy, as shown in Appendix A.1.

**Lattice of $\mathcal{S}$ (operations)**. We use the hierarchical structures (i.e., layered titles and subtitles) from the OAuth authorization pages to construct this lattice. Intuitively, we utilize operations from the lower layers of the hierarchical structure as explanations for those from the upper layers. For example, "change", "add" and "delete" are further explanations of the operation "access and edit" in the

layer above. Figure 4 (lower right) shows the extracted permissions from the OAuth prompt shown in Document 2 in Figure 2). Such layer relationship can be reflected by HTML tags (e.g., *<h>*). We recursively perform such explanation steps until all the layers of permissions have been covered in the prompted OAuth page. We assign the symbol ⊥ (as shown in Figure 4) to represent the unknown relationship between permissions (e.g., "edit" and "access") and to mark the end of the lattice construction.

*3.4.2 Detecting Excessive Permissions.* Based on the lattices, PFCon can check the consistency between the (OP, OB) pairs, for object and operation fields separately. The main challenge lies in determining whether two objects or operations refer to the same entity, as the service providers may use different terminologies. To overcome this, we enlist the LLM to assess the semantic similarity of these terms, as illustrated in Table 2.

**Object-level checking**. PFCon checks the object fields of the (OP, OB) pairs in $\mathcal{S}$ and $\mathcal{R}$. Intuitively, it checks whether the object in one of $\mathcal{S}$.(OP, OB) pairs is contained in the object of any $\mathcal{R}$.(OP, OB), i.e., $\exists$(OP, OB) ∈ $\mathcal{R}$ such that $\mathcal{S}$.OB ≤ $\mathcal{R}$.OB. PFCon augments the LLM with *context* to enhance semantic understanding of the terms. We

**Table 3: Permission excess overview**

| # Permission excess | 1 | 2 | 3 | 4 | 5 | 6 | 7 | 8 | 9 | 10 | 11 | 12 | 13 | 14 | 17 |
|---|---|---|---|---|---|---|---|---|---|---|---|---|---|---|---|
| # Services | 56 | 35 | 25 | 28 | 5 | 5 | 1 | 4 | 4 | 3 | 3 | 5 | 1 | 2 | 2 |

use the general service description provided for IFTTT, accessible in the functionality artifacts, as the context. Additionally, we integrate the hierarchical relations from the lattice system knowledge to construct the "reasoning" for the CoT, enabling the LLM to grasp subsumptive relationships and assess similarity based on this knowledge. PFCon traverses the object lattice from $\mathcal{S}$. For each object, it queries the LLM with the prompt listed in the *object-level checking* row in Table 2. When the LLM finds a semantically identical object (indicated by GPT-4 responding 'Yes'), the object is marked as "required".

**Operation-level checking**. When $\mathcal{S}.\text{OB} \preceq \mathcal{R}.\text{OB}$, PFCon proceeds to check the operation fields. It checks whether $\mathcal{S}.\text{OP}$ has a lower or equal privilege level compared to the $\text{OP}$ that has the highest level in $\mathcal{R}$ (denoted by $\mathcal{R}.\text{OP}_h$), i.e., $\mathcal{S}.\text{OP} \preceq \mathcal{R}.\text{OP}_h$. To handle the semantic similarity between ($\text{OP}, \text{OB}$) during this step, PFCon applies a methodology similar to the object-level checking and employs the prompt queries outlined in the *operation-level checking* row in Table 2.

## 4 EVALUATION

Aligning with previous study [9, 18, 19, 23, 32, 42], we implement PFCon and evaluate its performance in IFTTT, the most popular TAP. In our evaluation, we aim to answer the following research questions (RQs).

**RQ1**. What are the characteristics and prevalence of the permission excess issues in the real-world TAP?

**RQ2**. How effective and accurate is PFCon in detecting permission excess issues?

**RQ3**. What are the root causes (RCs) of permission excess issues?

### 4.1 Dataset

We crawl the IFTTT website to collect the evaluation dataset, as described in Section 3.2. Out of the over 700 services supported on IFTTT, we eventually obtain 427 for analysis, excluding no-English services, services that require no OAuth authorization (e.g., public news websites). We find that the services have significantly diverse numbers of permission sentences. Among them, 48.6% have only one or two permission sentences, 21.4% have three to four, 22.3% have five to ten, and only 7.7% have more than ten.

**Ethical considerations.** We anonymize all the services included in our dataset, and refer to them as anonymous 4-digits in the subsequent analyses unless stated otherwise. Note that we have responsibly disclosed the identified issues to the dedicated developers before presenting our findings.

### 4.2 RQ1: Permission Excess Landscape

*4.2.1 Permission Excess Prevalence.* We present an overview of the permission excess issues detected by PFCon in Table 3. Out of the 427 services, we find an astonishing 179 (41.9%) of them have permission excess issues. Among the services with violations, most (144) have one to four excessive permissions, while 13 services have over ten excessive permissions.

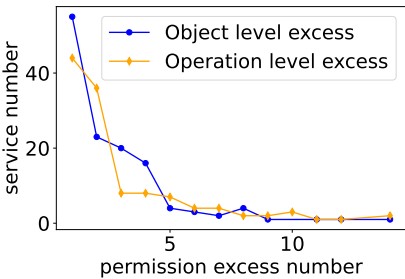

**Figure 5: Permission excess of service**

**Table 4: Features of permission excess: Top-5 operation/object detected as excessive.**

| Operation name | manage | modify | remove | create | write |
|---|---|---|---|---|---|
| Exceeded time | 93 | 84 | 52 | 29 | 15 |

| Object name | profile | setting | event | notice | country |
|---|---|---|---|---|---|
| Exceeded time | 15 | 8 | 8 | 7 | 7 |

**Table 5: User installation distribution for services with permission excess**

| User installation | 0-100 | 100-500 | 500-1k | 1k-5k | 5k+ |
|---|---|---|---|---|---|
| Service number | 76 | 29 | 17 | 13 | 44 |

**Permission excess prevalence by types**. As shown in Figure 5, 131 services have at least one object-level excessive permission, while 122 services have at least one operation-level excessive permission.

**Permission excess distribution**. From Table 5, we find that permission excess issues are more prevalent among services with fewer installations, as their developers might enforce less stringent checks for permission functionality consistency. We further investigate the root causes of such excessive permissions in RQ3 (Section 4.4).

*4.2.2 Characteristics of Permission Excess.* We further analyze permission features contributing to excess permissions, offering insights for developers to mitigate similar issues. Our study identifies both object and operation-level features, with the top five detailed in Table 4.

**Features of object-level excess**. Our findings indicate that the most commonly violated permissions pertain to profile information. Such excess permissions could potentially result in unauthorized access to sensitive user data, including contact information. In general, for individual users, highly personal objects are prone to be abused, such as videos, documents, tasks, messages, calendars, and contacts. Their unauthorized access and manipulation could incur significant privacy damage. For industry users, highly valuable objects are the primary targets, such as configuration, enterprise, message, voice command, subscription and product. Some objects, such as configuration and voice commands, may impact the physical devices and environments due to the involved smart home devices, hence also posing a safety risk.

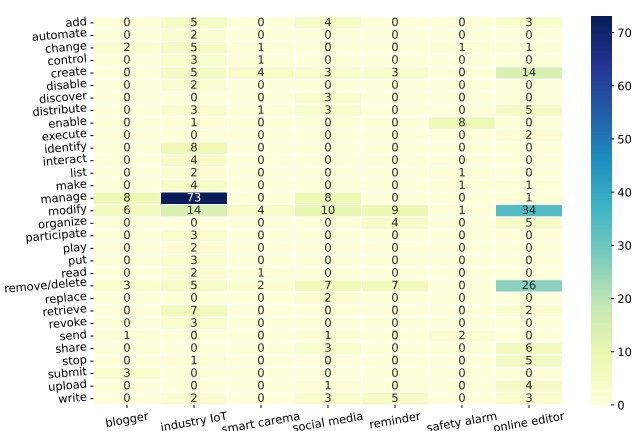

**Figure 6: Permission excess distribution among categories**

**Features of operation-level excess**. Generally, the most sensitive and dangerous operations include the *modify/change* and *remove/delete* operations since the user may lose data completely upon unauthorized execution. Our detection results also show that these operations are the most prevalent among the exceeded permissions. Specifically, the *modify* operation has 84 excessive occurrences, and *remove/delete* has 52 occurrences. However, these operations are only available in certain service categories like online editors (e.g., Google Sheets/Docs, and OneDrive) as shown in Figure 6. For example, the most prevalent excessive operation *manage* (93 excessive occurrences) is majorly present in the category of industry IoT (e.g., smart agriculture and company camera system). PFCON further identifies that the most permission excess reported are under these two categories, i.e., online editors and industry IoT.

## 4.3 RQ2: Performance Evaluation

In this RQ, we first present the overall detection accuracy, and then further investigate the step-wise accuracy of PFCON with the discussion on plausible causes for detection failure.

*4.3.1 Overall Accuracy.* We first construct a benchmark dataset, and evaluate PFCON using it.

**Benchmark dataset assembly**. Since there is no existing benchmark dataset to evaluate the accuracy of PFCON, we resort to manual effort to construct a suitable one. We first divide the dataset into four groups based on the number of permission sentences extracted from a service: group 1 with one to two permissions; group 2 with three to four; group 3 with five to nine; group 4 with ten or more. Then, we randomly and proportionally select 15% (i.e., 64 out of 427) of the services from our dataset to form the benchmark dataset that can evenly reflect the permission distribution across our dataset, including 30 from group 1, 15 from group 2, 15 from group 3, and 4 from group 4.

**Ground truth construction**. To construct the ground truth, we recruit three volunteers (one PhD and two master students) that major in computer science. To avoid personal bias of excessive permission perceptions [12, 22, 25], we create explicit labeling instructions that explain the task and provide some corner cases. We give the volunteers a detailed and objective tutorial with several

**Table 6: PFCON accuracy**

**(a) Benchmark performance**

| | Label | Detector | FP | TP | FN | TN | TPR | TNR |
|---|---|---|---|---|---|---|---|---|
| **POS** | 17 | 20 | 3 | 17 | 0 | 44 | 100% | 93.6% |
| **NEG** | 47 | 44 | | | | | | |

POS ($\mathcal{S} - \mathcal{R} \neq \emptyset$): permission excess, NEG ($\mathcal{S} - \mathcal{R} = \emptyset$): Non permission excess (see Definition 3).

**(b) PFCON stage accuracy**

| Accuracy | P1 | P2 | P3 | P4 |
|---|---|---|---|---|
| **Plain** | 40% | N/A | 65% | 85% |
| **Context** | 75% | N/A | 75% | 90% |
| **Context + CoT** | 95% | N/A | 95% | 100% |
| **Overall** | 94% | 100% | 96% | 94% |

P1: Prompt for task 1, P2: Prompt for task 2, P3: Prompt for task 3, P4: Prompt for task 4

case studies. To prevent any possible bias caused in this process, they are not asked to generate $\mathcal{S}$ or $\mathcal{R}$ representations. Instead, they look through each sentence in the OAuth page (e.g. Document ②of Figure 2), check whether it is needed by referring to descriptions (e.g., Document ① of Figure 2), and annotate unnecessary (OP, OB) pairs, e.g., (delete, files). The inter-annotator agreement quality measurement using Krippendorff's Alpha [30] achieves 0.823 and an alpha of 0.80 or higher indicates a good correlation between annotators [7]. To handle the disagreement, a security expert with 8 years (since 2015) experience is invited to discuss and verdict on the final label.

**Results**. Table 6 summarizes the performance of PFCON on the benchmark dataset. It achieves the True Positive Rate (TPR) of 100% and True Negative Rate (TNR) of 93.6% with only three false positives (FP). The detailed results for each service is available in Appendix A.2. We will further dicuss the possible cause in Section 4.3.3. PFCON also exhibits a great capability to capture the permission excess issues, with a recall of nearly 1. Even with a few false positives, PFCON remains highly effective as privacy analysts can easily confirm the results and rule them out. In particular, we find 17 (26%) out of the 64 benchmark services have permission excess issues. Three services have ten or more permission excess.

*4.3.2 Stage-wise Accuracy.* Considering the main methodology of our work consists of NLP tasks, we ensure the reliability of PFCON by checking the accuracy of each analysis stage, as listed in Table 6(b).

We compare three prompt patterns - plain, context information, and context information enhanced by CoT - using a small dataset. For task 2, we employ plain prompts without context and CoT, as GPT-4 can easily extract (OP, OB) pairs from simple sentences. With 20 queries per task, we manually evaluate the results. The context information + CoT prompt consistently performs the best. In task 3, GPT-4 effectively handle self-defined structures with Lattice System and CoT. For example, when working with service 3828, it understood and used service-specific structures, like "pot balance ≺ account", successfully captured by the lattice system.

Following validation on a small dataset, we proceed to evaluate on all services in our dataset, assessing stage-wise accuracy (last row of Table 6b). For each task, we employed a random selection and meticulous screening of 100 queries. PFCON consistently achieved high accuracy across all stages as shown in Table 6b.

*4.3.3 Case Analysis.* We investigate the false positive and permission-level miss detection (see Appendix A.3 for the detailed case studies). **Permission-level missed detection**. The reason for missing the excessive permissions in the service 2999, 6516 and 9650 arises from the difficulty in capturing the folder structure defined by the services. The service 9650 has a self-defined folder structure: workspace $\supseteq$ board $\supseteq$ lists $\supseteq$ card. Unfortunately, neither the $\mathcal{S}$ nor the $\mathcal{R}$ information provides any insights into this hierarchy. Consequently, our lattice system is unable to capture and construct this structure accordingly. The only action (i.e., "*Create a card*") requires updating an existing board, and the permission to *create boards* is thus excessive. PFCon misses detecting this permission since GPT-4 considers *board* and *card* are semantically similar. Service 2999 and 6516 have a similar issue.

## 4.4 RQ3: Root Causes for Permission Excess

After revealing that around one third of the services request excessive permissions, we further investigate their root causes to obtain an in-depth view on such issues. We randomly select and manually check 50 (out of 179) services with excessive permissions. By the root cause types, we group them into the following three categories. Besides 19 services with unidentifiable causes, we group the rest 31 into the following four categories.

**RC1: Permission bundle (9 services)**. When similar permissions are grouped into bundles, there is an increased risk of permission excess. More specifically, IFTTT has to apply for the whole group of permissions even if only part of them are required. This RC is the major cause for permission excess in popular services like 4682 (5.3k installation), 6750 (23.8k), and 0032 (114.k). After consulting IFTTT and service developers, they have confirmed this as a plausible cause for potential permission excess. Based on their types, permission bundle can be further divided into two categories: **bundle in operation** and **bundle in object**. For example, operations such as "access", "change", "add" and "delete" are commonly bundled, and objects such as "photos" and "videos" are typically bundled. We provide concrete case studies in Appendix A.3 for detailed reference.

**RC2: Defect Permission System Implementation (9 services)**. When developers lack a thorough understanding of a service's functionality and the nuances of privacy protection, there is a heightened risk of implementing excessive or even irrelevant permission requests. For example, a financial service 3828 (21.9k installation) implements a dangerous permission "Freeze and unfreeze your card" (highly sensitive operation) through IFTTT which is neither required nor necessary for its functionality.

**RC3: Ambiguous Permissions (8 services)**. When permission requests are ambiguously formulated, they can create disparities in downstream user understanding and perception. This lack of clarity may lead to more permissions requested than actually necessary for the service functionality, compared to the user interpretation of the request. For example, service 6460 (working together with IoT devices), has permission requests like "control your other smart home devices" which is ambiguous and vague compared with specific permissions ("turn on/off your devices").

**RC4: Template Usage (5 services)**. Different services belonging to the same company may share the same template or similar permission management patterns without considering the usage-specific scenarios. This could consequently introduce inaccurate and excessive permissions requested. For example, all services (including cleaning robots, coffee machines, dishwashers, dryers, etc.) under the same electronic company H (full name anonymized) share one OAuth template, without considering the customized and detailed service usage scenarios. This issue also exists among famous service providers like 6750 (23.8k installation).

## 5 RELATED WORK

PFCon is related to the privacy and security of IoT integration. In this section, we summarize existing studies related to them. We present a brief comparison between PFCon and other previous work on TAP in Appendix A.4.

**Privacy/Security in the trigger-action service**. Bastys et al. [13] identify the security issues inherent in the TAP and provide mitigations. Fernandes et al. [23] provide a protection mechanism to safeguard the applet execution. Yunang et al. [18, 19] design the data minimization to reduce the data attributes transferred to the IFTTT. In terms of the privacy and security analysis, PFCon confirms the privacy/security existence [14, 35] in TAP and complements these studies with the study of permission-level excess.

**Automatic testing of IoT integration**. HomeScan [14, 33] proposes a model-checking tool for smart homes and verifies safety and security properties. AutoTap [42] and AutomatedLTL [44] provides a checking method based on the LTL formula to detect security violation in applets and provide suggestion for users. TAIFU [32] tests trigger-action service like IFTTT and finds many violations against GDPR. PFCon's approach of auto-connecting services is inspired by them. In summary, most existing work focuses on applet-level testing, while PFCon targets the service-level permission analysis.

**NLP based security scrutinizing**. PolicyLint [10] utilizes sentence-level NLP to analyze data collection and sharing in privacy policies, identifying nine types of semantic contradictions. However, it cannot compare stated privacy claims with actual behavior. PoliCheck [11] applies PolicyLint and AppCensus [6] to detect the inconsistency for android app but requires available source code. GUILeak [40] uses annotated dataset and share similar ideas with PoliCheck. All of the current research work [26, 27] cannot handle platforms like TAI whose permission is not unified. In general, PFCon targets permission excess detection with various permission eco-systems and limited text corpus. PFCon can achieve good performance even without source code and user behavior or logging analysis.

## 6 CONCLUSION

In this work, we develop PFCon to examine permission-functionality consistency among implementations of TAI's services. PFCon is capable of first extracting the required permissions according to the functional capabilities provided by a TAI service. It then subsequently performs a consistency check to ensure that only the necessary permissions are requested during the service runtime. Through our systematic evaluation, we have identified nearly a third of the IFTTT services have been requesting excessive permissions, marking the alarming importance of responsibly enforcing the principle of least privilege in practice.

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

## Table 7: Expanded keywords for PolicyLint

| Type | Keyword |
|---|---|
| with | along with, together with, accompany, in addition to |
| contain | composed of, containing |

## Table 8: Details of Benchmark

| Service Name | B [†] | G | D | Service Name | B | G | D |
|---|---|---|---|---|---|---|---|
| 7858 | 0 | 0 | 0 | 9393 | 0 | 0 | 0 |
| 8060 | 0 | 0 | 0 | 1793 | 0 | 0 | 0 |
| 3814 | 0 | 0 | 0 | 3911 | 0 | 0 | 0 |
| 4591 | 0 | 0 | 0 | 5902 | 0 | 0 | 0 |
| 6853 | 0 | 0 | 0 | 2977 | 0 | 0 | 0 |
| 7078 | 0 | 0 | 0 | 2453 | 0 | 0 | 0 |
| 6748 | 0 | 0 | 0 | 9665 | 0 | 0 | 0 |
| 6451 | 0 | 0 | 0 | 7470 | 0 | 0 | 0 |
| 5152 | 0 | 0 | 0 | 1645 | 0 | 0 | 0 |
| 9366 | 0 | 0 | 0 | 2945 | 0 | 0 | 0 |
| 1328 | 0 | 0 | 0 | 1535 | 0 | 0 | 0 |
| 4202 | 0 | 0 | 0 | 1020 | 0 | 0 | 0 |
| 1120 | 0 | 0 | 0 | 6151 | 0 | 0 | 4 |
| 0004 | 0 | 0 | 0 | 0069 | 0 | 0 | 1 |
| 7008 | 0 | 0 | 0 | 0553 | 0 | 0 | 4 |
| 2869 | 0 | 0 | 0 | 3612 | 4 | 4 | 4 |
| 8108 | 0 | 0 | 0 | 8027 | 1 | 1 | 1 |
| 9970 | 0 | 0 | 0 | 0047 | 1 | 1 | 3 |
| 9700 | 0 | 0 | 0 | 0042 | 2 | 2 | 2 |
| 6860 | 0 | 0 | 0 | 6188 | 1 | 1 | 4 |
| 5590 | 0 | 0 | 0 | 0021 | 6 | 6 | 8 |
| 2534 | 0 | 0 | 0 | 0031 | 3 | 3 | 3 |
| 9378 | 0 | 0 | 0 | 0308 | 5 | 5 | 5 |
| 9984 | 0 | 0 | 0 | 1406 | 1 | 1 | 3 |
| 3970 | 0 | 0 | 0 | 9352 | 8 | 8 | 8 |
| 6751 | 0 | 0 | 0 | 4682 | 13 | 13 | 13 |
| 0383 | 0 | 0 | 0 | 8624 | 1 | 1 | 1 |
| 9459 | 0 | 0 | 0 | 2999 | 6 | 6 | 9 |
| 5243 | 0 | 0 | 0 | 3811 | 2 | 2 | 6 |
| 9009 | 0 | 0 | 0 | 9650 | 9 | 11 | 13 |
| 9124 | 0 | 0 | 0 | 6155 | 10 | 10 | 12 |
| 7718 | 0 | 0 | 0 | 6516 | 2 | 3 | 5 |

[†] G: GroundTruth, D: Detector, B: G∩D

# A  APPENDIX

## A.1  Expanded keywords

We list the expanded keywords as a complementary for PolicyLint in Table 7.

## A.2  Details of benchmark

We present a detailed view of our benchmark in Table 8.

## A.3  Case Study

**Case study 1.** Service 4682 bundles different operations "access", "change", "add" and "delete" (your documents) into one group, IFTTT developers acknowledge that "delete" is not required but they have to apply based on the permission bundle setting of 4682. Service 6432, 6750 and 7804 also bundle "create", "modify" and "delete" (your documents) into one group. Similarly, different objects: "photos" and "videos" are bundled and have to be applied together regardless of actual requirement.

## Table 9: A comparison among PFCon and other studies

| | Service-level | | Applet-level | |
|---|---|---|---|---|
| | Operation | Object | Data-flow | Chaining |
| TAIFU [32] | | | ✓ | |
| MinTap [18] | | | ✓ | |
| LazyTap [9] | | | ✓ | |
| ETAP [19] | | | ✓ | |
| MEDIC [14] | | | | ✓ |
| DTAP [23] | | | | ✓ |
| TKPERM [36] | | | | ✓ |
| AutoTap [42] | | | ✓ | ✓ |
| AutomatedLTL [44] | | | ✓ | ✓ |
| PFCon | ✓ | ✓ | | |

**Case study 2.** Template example:

> The template content:
> 1. Identify your home appliances.
> 2. Access to your dryer (service name).
> 3. Monitor appliance.
> 4. Control appliance.
> 5. Get and modify settings.
> 6. Forward events.

In this situation, the service cookit and cooktop don't fit in the template and are detected with permission excess issues. Template usage also happens to another company GE that provides smart home services like water heater, washer, oven, refrigerator, dryer, air conditioner and dishwasher.

Company G has integrated multiple services: G calendar, G assistant, G contacts, G docs, G drive, G sheets, G tasks and G wifi into IFTTT. G docs, drive and sheets share very similar pattern and all contain requested permission "Share and stop sharing your G Drive files (G Docs documents/spreadsheets) with others", but this permission is not required by IFTTT. Even worse, all services (i.e. cleaning robot , coffee machine, dishwasher, dryer etc) under the company H share one OAuth template which only changes the service name.

**Case study 3.** Service 6460 only exposes trigger apis (read permission required) for IFTTT, but "control your other smart home devices" (read and action permission) is requested. This happens frequently for IoT devices due to the poor organized permission system, developers is unable to provide more fine-grained permission compared with Software like Android or Cloud Storage. Service 3828 (one famous online banking) provides "Freeze and unfreeze your card" (highly sensitive operation) for IFTTT but actually such dangerous permission are not required. From this case study, we observe that even financial company would bring in such security issue. Considering its severe effects being manipulated, this must raise an alert and be handled properly by both IFTTT and service providers.

## A.4  Comparison with other work

The comparison with reference to other work is listed in Table 9.

