# OpenReview forum: "Don’t bite off more than you can chew: Investigating Excessive Permission Requests in Trigger-Action Integrations"
_ACM.org/TheWebConf/2024/Conference — TheWebConf24 Oral_

### Official Review · Reviewer_wLAG · 2023-11-06

**Novelty:** 6
**Technical Quality:** 6

**Review:**

### Summary:
Using the newly proposed approach PFCon, the paper studies security and privacy issues in the context of trigger action platforms (TAPs). In particular, the paper considers issues that result from an excess of object and operational-level permissions. PFCon utilizes LLM prompts to identify the requested and required permissions to subsequently model their relationship using a lattice-based approach. For the study, the paper considers 427 services that are offered on the IFTTT website and discovers significant issues in one-third of the studied services. To validate and confirm the accuracy of their approach, the authors conduct a small user study (involving people with a computer science background), where the participants manually classify 64 services.

### Pros:
+1: Very well-written and presented paper

+2: Quite the interesting and novel approach to tackle the outlined challenges

### Cons:
-1: Some minor issues/questions remain

The paper is a very interesting read and nicely outlines the problem, the design, and the evaluation, including the results and potential reasons for the observed situation. From my point of view, I am not able to identify significant flaws in the work, which is why I would like to see it at the main conference. I also like the extra content in the appendix of the paper. Below, I will attach a list of minor comments that could allow the authors to further improve their paper before publication.

### Detailed Comments:

#### -1: Minor issues (sorted by occurrence)
- Unfortunately, the paper does not indicate whether PFCon will be publicly available (open-sourced). Should third parties be allowed to run the proposed approach?
- My impression is that Section 3.2 offers quite a lot of engineering details that are not required to understand the design of PFCon. As a result, I would recommend the authors to move most of the details regarding the identification of login fields, etc., to the appendix of the paper and use the space for more important content.
- The paper outlines that 700 services are available and that 427 have been selected for the evaluation. While the paper gives a few pointers as to why the services are considered in the evaluation, the corresponding paragraph is quite vague. Are foreign languages and the lack of OAuth authentication the only reasons for exclusion (what is the distribution)? Or are there any other reasons at play that the number is reduced by quite a lot?
- The authors state that they reached out to the service developers to inform them about their findings. Unfortunately, the paper does not report whether any changes have been made and how many responses the authors received. I would like to see additional details in this regard, possibly as part of the appendix. Moreover, it would be interesting to observe how the situation evolves now that PFCon is available and allows for repeated studies.
- The order of Tables 4 and 5 is different from their first reference in the text. I would recommend swapping them.
- The conclusion of the paper is rather short. I hope that the authors can add a few more details to this part of the paper once they have condensed Section 3.2 (see above).
- The text embedding of the tables in the appendix is quite brief. I would like to have some more elaborate context for each of the subsections. Since the number of pages for the appendix is not limited by the submission requirements, the authors can easily extend it.
- Picking up on the previous comment, the semantics of Table 9 are not pointed out in the current version of the paper. What is the rationale behind the grouping of different approaches?

#### Nits:
- There is a typo in Figure 6: "carema" should be "camera"
- There are a few lines that exceed the column width, for example, in Sections 4.3.3 and 5.
- I believe Android should be capitalized in Section 5. Moreover, I think the plural of "app" is needed in the same sentence.
- The first sentence of Section 6 most likely sounds better if it is written in past tense.

### Post-Rebuttal

I kindly thank the authors for responding to the reviews and outlining their proposed changes.
After these comments, I do not have any follow-up questions concerning the aspects that I initially raised as part of my review.
I am curious to see whether the authors still find the time to respond to the response by Reviewer mciy.
Certainly, the approach is "flawed" in the sense that it can only work with the textual information that is available.
Personally, I would not discredit the proposed approach for this reason because effectively any design that relies on the textual information is impacted/limited in the same way.

**Questions:**

n/a

**Reviewer Confidence:**

3: The reviewer is confident but not certain that the evaluation is correct

**Scope:**

4: The work is relevant to the Web and to the track, and is of broad interest to the community

---

### Official Review · Reviewer_hRwH · 2023-11-12

**Novelty:** 6
**Technical Quality:** 6

**Review:**

### Summary

In this paper, the authors develop a tool, named PFCon, which is dedicated to detect the permission-functionality consistency issue in trigger-action integrations. Specifically, the consistency issue is divided into object level and operation level. Thus, the authors firstly collect TAIs from IFTTT. Then, taking advantage of ChatGPT, they extract the required and requested permissions for each TAI. Last, they utilize a lattice system to identify excessive permission requests. The evaluation depicts the landscape of this issue in IFTTT, and conducts a large-scale analysis.

### Strength

- This story of this paper is quite complete, the motivation, methodology, and evaluation are organized logically and fluently.
- The authors reveal the permission excess issues in TAIs, which are widely-spread according to their experimental results. It is interesting.
- The adopted methodology is systematic, and the evaluation is comprehensive.

### Weakness

- The adopted lattice system needs to be better illustrated.
- Some key parts are missing or should be paid more attention. For example, there is no “threats to validity” before the conclusion.

### Comments

First of all, I think this paper is quite complete and deserves to be published. However, some revisions should be conducted before that. I will detail some concrete concerns in the following.

In Fig. 2, the authors should add sub-captions to clarify the requested permissions and required permissions. This will increase the readability of this figure.

One of the main concerns is the adopted lattice system. Firstly, in Section 2.3, the authors define $S$ and $R$ as requested and required permissions, respectively. Thus, can I simply extract the abused permission as the $p$, which complies $\exists p \in S. p \notin R$? Moreover, the description in Section 3.4 is unclear. For example, in Section 3.4.1, the authors detail the built object lattice and operation lattice. How are these two lattices built? Through ChatGPT? In Fig. 6, why adopts *bottom* instead of *top* as the default symbol? In Section 3.4.2, in object-level detecting, the authors say $\exists (OP, OB) \in R$ such that $S.OB \preceq R.OB$. According to the example, it only illustrates the *comment* object is on the same level of other required permission. Can you revise the example to illustrate the $\prec$ relation? In operation-level detecting, why does the tool proceed to check the operation fields after successfully detecting object-level permissions abuse? Moreover, I think a universal quantifier is needed before the $S.OP \preceq R.OP$. Last but not least, if you have constructed a lattice system, why do you still perform the comparison with the help of ChatGPT instead of directly performing the comparison, where ChatGPT cannot guarantee 100% precision on such a kind of task.

In Section 4.1, there is a brief description about the ethical considerations. However, I think more details should be discussed here. For example, how long have you taken to disclose the corresponding permission abuse to IFTTT and service providers after you discovered them. How many of them have been recognized and patched timely?

In the evaluation part, I prefer to see a separate paragraph after each RQ to summarize the findings of the current RQ.

In Table 6(a), the positive cases are defined as the $S-R$, which does not distinguish the object and operation level permission excess. On the one hand, if they are not distinguished, is it necessary to distinguish them in the methodology part? On the other hand, I would like to see a break-down analysis here. That is, how many cases are suffered object level permission abuse, as well as the operation level abuse.

What is task 1 to 4 in Table 6(b)? The authors should clarify this.

Last but not least, the authors should add an explicit paragraph to discuss the threats to validity.

**Questions:**

Please refer to the `Review` part.

**Ethics Review Description:**

The authors have addressed part of the ethical considerations, however, I think it should be discussed more in detail.

**Ethics Review Flag:**

Yes

**Reviewer Confidence:**

3: The reviewer is confident but not certain that the evaluation is correct

**Scope:**

3: The work is somewhat relevant to the Web and to the track, and is of narrow interest to a sub-community

---

### Official Review · Reviewer_kKe4 · 2023-11-23

**Novelty:** 3
**Technical Quality:** 3

**Review:**

----------- Summary -----------

The paper investigates the cross-entity permission management in TAIs, and tries to address the root causes of the applet-level security and privacy issues that have been the focus of the literature in this area. They develop PFCon, which extracts the required permissions based on all functionalities offered by an entity, and checks the consistency between the required and requested permissions on users’ assets. The study reveals some interesting findings such that nearly one third of the services
in these integrations request excessive permissions.

----------- Strengths -----------

S1. The paper is well written, and the logic is clear.

S2.  The authors conduct a large-scale study to study the permission excess issues.

S3.  The paper proposes an assessment approach to automatically identify permission excess issues from TAIs.

----------- Weaknesses -----------

W1. The motivation is not clear. It is unclear how serious is the permission excess problem in TAIs.

W2. It is not clear what resources (services) the paper used to collect the artifact. How are the services selected? What are the statistics of the collected information?

W3. The rationale of methodology design is lacking. For example, how and why is TAIFU selected as the analyzer? Similarly, how is GPT-4 model selected and used? How are the prompts designed?

W4. It is unclear why the two permission lattice systems (object lattice and operation lattice) are effective in capturing the permissions. How would the performance of this step affect the result of the detection of excessive permission?

W5. The sample size is too small (64 samples).

W6. I also doubt about the novelty of PFCon, as it seems to combine the existing tools. The authors may want to clarify the innovation of the approach clearly.

**Questions:**

-	What is the motivation of the work? Why solving the permission excess problem is important?
-	What are the resources (services) the paper used to collect the artifact? How are the services selected? What are the statistics of the collected information?
-	What is the rationale of methodology design. For example, how and why is TAIFU selected as the analyzer? Similarly, how is GPT-4 model selected and used? How are the prompts designed?
-	Why the two permission lattice systems (object lattice and operation lattice) are effective in capturing the permissions? How would the performance of this step affect the result of the detection of excessive permission?
-	What is the novelty of PFCon, as it seems to combine the existing tools?

**Reviewer Confidence:**

4: The reviewer is certain that the evaluation is correct and very familiar with the relevant literature

**Scope:**

4: The work is relevant to the Web and to the track, and is of broad interest to the community

---

### Official Review · Reviewer_mciy · 2023-11-24

**Novelty:** 3
**Technical Quality:** 4

**Review:**

Summary
The authors study the cross-entity permission management in the trigger-action integrations (TAIs). More specifically, they propose and develop a prototype named PFCON to analyze the permission-functionality consistency in TAIs. In detail, PFCON employs a large language model to analyze the required permissions based on functionalities offered by an entity and checks the consistency between the required and requested permissions on users’ assets. Applying PFCON to TAIs built upon IFTTT, the authors find several TAIs request excessive permissions.

Strengths
- The authors take the first step to investigate the problem of excessive permission requests in TAIs and develop a prototype named PFCON to detect the problem.
- The authors do find problematic cases in real-world TAIs.

Weaknesses
- PFCON does not take applets (i.e., code) into analysis, making the results unreliable.
- The feedback on the discovered problematic cases is not disclosed.
- The source code of the prototype and the dataset have not been released.

Detailed comments
The paper studies an important problem in TAIs. However, I have the following suggestions and comments on the current version of the paper.

1. Clarifying the correctness of the proposed approach.
PFCON mainly analyzes the documentation of a TAI’s interfaces and functionalities and the corresponding description of the authorization page to identify excessive permission requests. My main concern is whether such textual information is reliable. If the textual information is incorrect, the analysis results of PFCON become unreliable. In my view, it is common that the textual information is incorrect or imprecise. For example, in Figure 2, can “tag” denote “name”? Can the “comments” refer to “modified_time” and/or “modified_by”? It seems that there is no ground truth about the corresponding relationship between different descriptions. Such a problem makes me worry about the correctness of the proposed approach.

2. Clarifying why PFCON does not analyze applets.
In my view, code is more reliable than textual information. I think PFCON can also perform code analysis on applets to verify the correctness of the results by purely analyzing the textual information. If the authors do not think so, provide a corresponding discussion in the paper.

3. Disclosing the detected problematic TAIs to IFTTT and including the feedback in the paper.
The authors should follow the responsible disclosure policy to disclose their findings to IFTTT. If the IFTTT can help confirm the correctness of the detection results, the authors could add feedback to the paper to demonstrate the effectiveness of PFCON.

4. Releasing the source code and dataset.
Will the authors release the source code of PFCON and the dataset used in the evaluation?

**Questions:**

1. Are the textual information about TAI’s interfaces and functionalities and authorization pages reliable?
2. Will the authors disclose their findings to IFTTT?

**Reviewer Confidence:**

3: The reviewer is confident but not certain that the evaluation is correct

**Scope:**

3: The work is somewhat relevant to the Web and to the track, and is of narrow interest to a sub-community

---

### Decision · Program_Chairs · 2024-01-22

**Decision:**

Accept (Oral)

**Comment:**

The paper presents a framework to discover when services on so-called trigger-action platforms such as IFTTT ask for more permissions than they need. Authors find some problematic cases in IFTTT, which were also responsibly disclosed. Detailed reviews raised several questions to which the authors provide suitable answers, and also make several promises for improving clarity in writing and adding additional details. All these should be implemented in the next version submitted.